# Effect of Carrot Callus Cells on the Mechanical, Rheological, and Sensory Properties of Hydrogels Based on Xanthan and Konjac Gums

**DOI:** 10.3390/gels10120771

**Published:** 2024-11-27

**Authors:** Elena Günter, Oxana Popeyko, Fedor Vityazev, Natalia Zueva, Inga Velskaya, Sergey Popov

**Affiliations:** Institute of Physiology of Federal Research Centre “Komi Science Centre of the Urals Branch of the Russian Academy of Sciences”, 50, Pervomaiskaya Str., 167982 Syktyvkar, Russia; gunter-ea@mail.ru (E.G.);

**Keywords:** xanthan gum, konjac gum, callus, cells, hydrogel, mechanical properties, texture, rheology, sensory properties

## Abstract

The study aims to develop a plant-based food gel with a unique texture using callus cells and a mixture of xanthan (X) and konjac (K) gums. The effect of encapsulation of carrot callus cells (0.1 and 0.2 g/mL) on properties of X-K hydrogels was studied using the mechanical and rheological analysis with a one-way ANOVA and Student’s *t*-test used for statistical analysis. Hedonic evaluation and textural features were obtained from 35 volunteers using a nine-point hedonic scale and a 100 mm visual analog scale with the Friedman’s test and the Durbin post hoc test used for statistical analysis. Mechanical hardness, gumminess, and elasticity increased by 1.1–1.3 and 1.1–1.8 times as a result of encapsulation 0.1 and 0.2 g/mL cells, respectively. The addition of cells to the hydrogels resulted in an increase in the complex viscosity, strength, and number of linkages in the gel. The hydrogel samples received identical ratings for overall and consistency liking, as well as taste, aroma, and texture features. However, the callus cell-containing hydrogel had a graininess score that was 82% higher than the callus cell-free hydrogel. The obtained hydrogels based on gums and immobilized carrot callus cells with unique textures may be useful for the development of diverse food textures and the production of innovative functional foods.

## 1. Introduction

Polysaccharides are widely used in the food industry due to their gelling, thickening, and emulsifying properties [1]. One commercial polysaccharide that is widely used in food systems is xanthan gum. Xanthan gum is used as a rheology modifier and food additive [2]. It has pseudoplastic rheological properties and temperature stability and is capable of stabilizing emulsions [1,2]. Xanthan gum is capable of synergistic interaction with konjac gum, which leads to the formation of a physical hydrogel [3]. Much research has been performed to understand the factors that regulate these interactions. As gelling systems, they are a subject of study for food science because they are good texturizing agents [4].

Xanthan gum is a microbial exopolysaccharide produced by the fermentation of glucose, sucrose, or lactose by the bacterium *Xanthomnas campestris* [5]. The polysaccharide backbone is composed of β-D-glucopyranose units linked by 1,4-glycosidic bonds [6,7]. The side chains are trisaccharides consisting of a D-glucuronic acid unit between two D-mannose units and are attached to the C-3 atom of every second glucose residue. About half of the terminal mannose residues contain a pyruvate group, the amount of which determines the viscosity of aqueous xanthan solutions [5]. The hydroxyl at C-6 of the non-terminal mannose residue is replaced by an acetyl group. The negatively charged side chains are wound back around the backbone, forming a helical structure supported by hydrogen bonds and protecting the backbone. Due to this, xanthan exhibits good resistance to degradation [7].

Konjac gum, found in the tubers of the *Amorphophallus konjac* plant, is a glucomannan consisting of a linear backbone of 1,4-linked mannopyranose units that are interspersed with linear 1,4-linked glucopyranose units [8]. Typically, the glucose to mannose molar ratio is 1:1.6 [8,9]. Acetyl groups replacing the hydroxyls at C-6 are present randomly along the chains [8,9]. Konjac gum is a biocompatible and biodegradable polysaccharide and is therefore often used in the food industry as a gelling agent or thickener [10]. However, the use of konjac gum is limited by its weak mechanical properties exhibited in aqueous solutions. Therefore, xanthan gum is used to increase the mechanical strength of the hydrogel [10]. Xanthan gum, combined with konjac gum or agar, is used in edible 3D inks [11,12].

Xanthan and konjac polysaccharides tend to gel depending on concentration and temperature conditions. In particular, xanthan forms “weak gels” in aqueous solutions at concentrations above 2%, while konjac is capable of forming stable gels at the same concentrations in the presence of alkali [4]. A synergistic interaction (gelling effect) occurs between xanthan and konjac glucomannan in the solution and in the gel phase [4]. It is assumed that interactions occur between the unsubstituted regions of the glucomannan backbone and the side chains of xanthan and that the ordered parts of the macromolecular complex act as junction zones in the gel phase [4]. Moreover, direct interaction occurs between the konjac (glucomannan) and xanthan (glucan) backbones if the latter is in a disordered conformation [8]. In addition, if xanthan molecules are in an ordered conformation, then their conformation can be changed to the one required for effective binding [8]. The unbranched konjac chains intertwine with the xanthan chains through hydrogen bonds and van der Waals forces. As a result, a three-dimensional grid structure is formed [13]. Van der Waals interactions, as well as hydrogen bonds, determine the formation of the spatial structure of biological macromolecules. Van der Waals interactions consist of weak electromagnetic interactions. A hydrogen bond is a form of association between an electronegative atom and a hydrogen atom covalently bonded to another electronegative atom. The OH groups in polysaccharides can form hydrogen bonds with neighboring hydroxyl groups. In particular, synergistic interactions between konjac-mannan and xanthan are noted through linkages between successive glucose units of konjac and xanthan, and these linkages depend on the presence of successive glucose units in the chemical structure of both gums [9].

Currently, the development of new technologies for the production of healthy plant-based food products is relevant due to the growing population of the planet, which must be provided with plant food [14]. One of these new technologies is the use of plant cell and tissue cultures to obtain new plant-based foods [14,15,16,17,18,19,20,21]. Both whole tissue cultures [14] and individual cells and tissues encapsulated in hydrogels can be used as a source of plant products [15,16,17,18,19,20,21]. In recent years, the method of 3D bioprinting of food hydrogels based on alginate, carrageenan, pectin, and agar with isolated plant cells or with plant cell and tissue cultures has been increasingly used [15,16,17,18,19,20,21]. Hydrogels with encapsulated callus cells represent great potential for creating a variety of food textures and producing innovative food products. This is due to the fact that callus cells have turgor pressure, and their immobilization in hydrogels gives the latter a unique texture of artificial plant tissues [16,17]. By adapting cells to environmental conditions, they can maintain viability and increase productivity [15]. In this case, living cells can form growing clusters, which in turn impart new textural, rheological, and mechanical properties to hydrogels.

To develop new methods for producing unique plant-based food items based on plant cell and tissue cultures, it is vital to investigate how the immobilization of callus cells in hydrogels impacts their mechanical and rheological properties. A change in the concentration of callus cells in a hydrogel does not clearly affect its mechanical and rheological properties and depends on both the composition of the hydrogel and the cells used. It has been previously shown that the addition of cells and an increase in the concentration of encapsulated lettuce leaf cells in a low-methoxylated pectin gel reduced the mechanical strength [22]. Mechanical and rheological properties decreased with an increase in the cell content in hydrogels from low-methylesterified tansy pectin, whereas these properties increased in hydrogels based on citrus pectin with a higher degree of methylesterification [23]. It was shown that the values of G′, G″, and hardness could both increase and decrease with an increase in the concentration of thawed lupine callus tissue cells in κ-carrageenan gels, which depended on the morphological features of the cells of different callus lines [20].

In this study, xanthan and konjac glucomannan, which have a significant synergistic impact, were combined with carrot callus cells to create composite hydrogels. It was suggested that the immobilization of carrot callus culture cells in the xanthan and konjac glucomannan hydrogels will allow obtaining hydrogels with unique structural–mechanical properties. The physical properties of food gels determine their mechanical behavior during chewing and are therefore critical parameters for the sensory perception of food [24]. Because modifying the physical properties of a food gel might affect its overall acceptability, hedonic and sensory evaluation is an important phase in the development of food gel loaded with callus cells [25]. Commonly examined acceptability aspects include overall liking, whereas relevant textural features may be classified into three domains, including mechanical, geometric, and moisture- and fat-related surface properties [26]. We examined ‘hardness’, ‘springiness’, and ‘adhesiveness’ as primary, and ‘gumminess’ as a secondary mechanical parameter. ‘Moisture’ and ‘easy to swallow’ subjective ratings were chosen to characterize moisture properties, which are important for characterizing the readiness of the bolus for swallowing. Graininess, as a geometric characteristic, was assumed to change when perceiving a gel containing callus cells that can form aggregates.

The study aims to develop a plant-based food gel with a unique texture using callus cells and a mixture of xanthan and konjac gums. For this, carrot callus cells were encapsulated into xanthan and konjac hydrogel, and the mechanical, rheological, and sensory properties of the final product were studied.

## 2. Results and Discussion

### 2.1. Mechanical Properties of Gum Hydrogels

Hydrogels were obtained based on an aqueous mixture of two types of gum, xanthan (X) and konjac (K), at various concentrations and component ratios. With an increase in the total concentration of gums from 0.5 to 1.0% at a constant ratio of xanthan and konjac gums as 1:1, the hardness, cohesiveness, gumminess, elasticity, and chewiness of hydrogels increased by 10.6, 1.1, 12.1, 3.6, and 7.3 times, respectively (Figure 1a–d,f). A subsequent increase in the gum concentration to 1.5% did not affect the specified mechanical properties of the hydrogels, with the exception of a decrease in elasticity (Figure 1d). At the same time, the springiness of the hydrogels decreased by 1.7 times with an increase in the total gum concentration from 0.5 to 1.5% at a X:K ratio of 1:1 (Figure 1e).

With an increase in the total gum concentration from 0.5 to 1.5% at a constant X:K ratio of 2:1, the values of hardness, cohesiveness, gumminess, elasticity, and chewiness increased proportionally to the gum concentration by 13.4, 1.2, 16.3, 1.8, and 15.2 times, respectively (Figure 1a–d,f). At the same time, the springiness of the hydrogels decreased by 1.1 times (Figure 1e). Previously it was also shown that when konjac and xanthan gums were mixed, the gel strength depended on the gum concentration. At higher gum concentrations, the gel strength increased [11]. Increasing the content of xanthan and konjac gums led to the formation of a denser network structure and improved mechanical properties [27]. Hardness and chewiness increased with increasing concentration of κ-carrageenan-konjac gum mixed gels [28]. At the same time, the concentration of mixed gels did not have a significant effect on springiness and adhesiveness. The network structure of the gel became denser with increasing gel concentration from 1 to 3%, which corresponded to an increase in chewiness, hardness, and resilience [28].

At a constant X:K ratio of 3:1, the hardness, gumminess, and chewiness of the hydrogels increased by 3.3 times with an increase in the total gum concentration to 1.0%, whereas with a further increase in the latter, the indicated mechanical characteristics did not change (Figure 1a,c,f). Cohesiveness and springiness increased slightly only at the highest gum concentration (Figure 1b,e). Elasticity increased by 1.3 times only at a gum concentration of 1.0% (Figure 1d).

An increase in the X:K ratio from 1:1 to 3:1 at a constant total gum concentration of 0.5% led to a proportional increase in hardness, gumminess, elasticity, and chewiness by 7.3, 7.8, 3.7, and 4.6 times, respectively (Figure 1a,c,d,f). In this case, cohesiveness and springiness either did not change or decreased by 1.7 times, respectively (Figure 1b,e).

Increasing the X:K ratio from 1:1 to 2:1 at a constant total gum concentration of 1.0% did not affect the mechanical characteristics except for a slight decrease in elasticity. At the same time, increasing the X:K ratio to 3:1 resulted in an increase in hardness, gumminess, elasticity, and chewiness by 2.3, 2.1, 1.3, and 2.0 times, respectively (Figure 1a,c,d,f). The values of cohesiveness and springiness either did not change or slightly decreased, respectively (Figure 1b,e).

At a constant total gum concentration of 1.5%, the highest values of hardness, gumminess, elasticity, and chewiness were observed at a X:K ratio of 2:1 (Figure 1a,c,d,f). Cohesiveness increased slightly only at a X:K equal to 3:1, and springiness decreased slightly only at a X:K ratio of 2:1 (Figure 1b,e).

The ratio of xanthan and konjac gums and their total concentration had a greater effect on the mechanical properties of the hydrogels, such as hardness, gumminess, elasticity, and chewiness. In general, an increase in the total gum concentration and an increase in the proportion of xanthan gum in the hydrogel contributed to the enhancement of these mechanical properties. The formation of a harder and more elastic gel may be due to synergistic interactions between xanthan and konjac [9]. In particular, interactions occur between the glucomannan backbone of konjac and the side chains of xanthan [4]. The unbranched konjac chains intertwine with the xanthan chains through hydrogen bonds and van der Waals forces [13]. The OH groups in polysaccharides can form hydrogen bonds with neighboring hydroxyl groups. The increase in the hardness of the hydrogels with an increase in the proportion of xanthan gum is probably due to the fact that a larger number of xanthan side chains, which are trisaccharides consisting of a D-glucuronic acid unit between two D-mannose units, interact with unsubstituted regions of the konjac glucomannan backbone [4]. In addition, an interaction may occur between the glucomannan backbone of konjac and the glucan backbone of xanthan, if the latter is in a disordered conformation [8]. In particular, the synergistic interactions between konjac and xanthan are formed through the bonds between successive glucose units of konjac and xanthan [9].

The data obtained are consistent with previously obtained data, which showed the formation of a stronger gel structure with a higher proportion of xanthan gum in binary mixtures of *Lepidium perfoliatum* seed gum and xanthan gum [1]. Adding xanthan gum to the agar/konjac glucomannan gel system led to an increase in gel strength [3]. Previously, it was shown that X:K in the ratio 3:1 (0.075%/0.225%) showed the best physicochemical, rheological, textural, and sensory properties [9].

In the next experiment, a total gum concentration of 1.0% and different X:K ratios (1:1, 2:1, and 3:1) in water were selected. Carrot callus cells were added to these hydrogels in an amount of 0.1 g/mL. This allowed us to identify how the addition of plant cells affects the mechanical characteristics of hydrogels at different X:K ratios, as well as how the X:K ratio affects the mechanical properties of gum/callus hydrogels. The addition of callus cells to hydrogels with X:K ratios of 1:1, 2:1, and 3:1 resulted in an increase in hardness, gumminess, elasticity, and chewiness by 1.7–2.4, 1.6–2.2, 1.3–1.4, and 1.6–2.2 times, respectively, and a decrease in cohesiveness by 1.1 times (Figure 2). The exception was the similar elasticity values for hydrogel samples with and without cells at a X:K equal to 3:1. The springiness of the hydrogels did not change with the addition of callus cells.

Increasing the hardness of hydrogels with the addition of callus cells may be related to the ion-exchange properties of the plant cell wall. In particular, the cell wall surface has a negative charge due to the presence of free carboxyl groups (COO-) of galacturonic acid residues in the cell wall pectins [29]. Due to the predominance of negative fixed charges in the cell wall, cations are concentrated on its surface. In turn, these cations can interact with the COO- groups of the glucuronic acid residues of the side chains of xanthan, which can lead to the formation of a harder hydrogel. In addition, it is possible that the increase in gum/callus hydrogel hardness was due to the emergence of a synergistic interaction of callus cell wall pectins with xanthan and konjac. A similar observation was previously made for sodium alginate interacting with a konjac glucomannan–xanthan gum complex [30]. Hydroxyl groups of xanthan can form intermolecular hydrogen bonds with carboxyl groups of alginate in composite granules [31]. The increase in the hardness of gum/callus hydrogels can also be attributed to the turgor pressure of callus cells and their own hardness, which leads to the formation of a unique texture of hydrogels [16,17].

Gumminess is defined as the product of hardness and cohesiveness [32], and similar to hardness, the gumminess values of hydrogels have increased. Chewiness refers to the amount of energy required to grind a material into an ingestible state [32]. The chewiness depends on the hardness and gumminess of the hydrogel; therefore, harder and stickier hydrogels had higher chewiness values. The cohesiveness expresses the strength of the internal bonds that support the body of the product [33]. The decrease in the cohesiveness of the hydrogels was likely due to a decrease in the strength of the internal bonds in the gum hydrogels when cells were encapsulated in them.

Mechanical characteristics were similar for gum/callus hydrogels at X:K ratios of 1:1 and 2:1 (Figure 2b). Increasing the X:K ratio to 3:1 caused an increase in hardness, gumminess, and chewiness by 1.5–1.6 times. At the same time, cohesiveness, elasticity, and springiness of the gum/callus hydrogels did not change.

To obtain gum/callus hydrogels, carrot callus cells (0.1 and 0.2 g/mL) were encapsulated in a hydrogel based on a 1.0% mixture of xanthan and konjac gums in a 2:1 ratio and carrot juice (Figure 3). This made it possible to obtain hydrogels with a unique texture of artificial plant tissues due to the turgor pressure of plant cells. Similar hydrogels free of callus cells served as a control.

Using instrumental texture analysis, it was found that the hardness, cohesiveness, gumminess, elasticity, springiness, and chewiness of cell-free gels (X2K1-J) based on gums (X:K ratio of 2:1) and carrot juice were 1.15 ± 0.32 N, 0.91 ± 0.04, 1.05 ± 0.28 mm, 3.16 ± 0.25, 0.94 ± 0.08, and 0.98 ± 0.25, respectively (Figure 4). The addition of callus cells of *Daucus carota* (DC) to carrot juice-based gum hydrogels (X2K1-0.1DC-J and X2K1-0.2DC-J), as well as to water-based ones, led to an increase in hardness, gumminess, elasticity, and chewiness by 1.3–1.8, 1.2–1.6, 1.1, and 1.2–1.5 times, respectively. In addition, there was a decrease in cohesiveness and springiness by 1.1–1.2 times at higher cell concentrations (X2K1-0.2DC-J). With an increase in the cell content in hydrogels from 0.1 to 0.2 g/mL, the values of hardness, gumminess, elasticity, and chewiness increased by 1.4, 1.3, 1.1, and 1.3 times, respectively (Figure 4). At the same time, the cohesiveness of hydrogels decreased by 1.1 times. The springiness of X2K1-0.1DC-J and X2K1-0.2DC-J hydrogels remained unchanged.

The improvement in the mechanical characteristics of the gum hydrogels with increasing cell concentration was probably due to the occurrence of a synergistic interaction between the pectin macromolecules of the callus cell walls and the xanthan and konjac macromolecules. In particular, the carboxyl groups of the galacturonic acid residues of pectin can form intermolecular hydrogen bonds with the hydroxyl groups on the main chain (β-1,4-glucan) and the side chains of the xanthan macromolecule. In addition, hydrogen bonds can be formed between the hydroxyl groups of pectin and the carboxyl groups of the glucuronic acid residues of the side chains of xanthan, as well as the carboxyl groups on the terminal side chains of xanthan. The carboxyl groups of the galacturonic acid residues of pectin can form intermolecular hydrogen bonds with the hydroxyl groups on the main chain of konjac glucomannan. With increasing cell concentration in the hydrogels from 0.1 to 0.2 g/mL, more callus cell wall pectins formed intermolecular hydrogen bonds with xanthan and konjac, resulting in the formation of a hydrogel with higher mechanical characteristics. Moreover, the increase in the mechanical characteristics of gum/callus hydrogels with increasing cell concentration can also be explained by the turgor pressure of callus cells and their own hardness, which both increase with increasing cell number.

Hydrogels based on a 1.0% mixture of xanthan and konjac gums with an inverted X:K ratio of 1:2 and carrot juice were prepared. The hardness, gumminess, elasticity, and chewiness of the cell-free and cell-containing hydrogels were 6.3–7.8, 5.4–7.7, 1.5–1.7, and 5.1–7.3 times lower, respectively, than those of the hydrogels prepared with a 2:1 mixture of xanthan and konjac (Figure 5). This indicates a significant decrease in the mechanical characteristics of hydrogels with a predominant proportion of konjac gum. The decrease in the hardness and other mechanical characteristics of hydrogels with a decrease in the proportion of xanthan gum is probably due to the fact that a smaller number of xanthan side chains interact with the unsubstituted regions of the konjac glucomannan backbone. This resulted in a weaker synergistic interaction between xanthan and konjac molecules and the formation of a weaker hydrogel. It has been previously shown that increasing the proportion of xanthan (X:K = 3:1) leads to an increase in the cohesiveness and gumminess of the polysaccharide mixture, which indicates the predominant effect of xanthan on enhancing mechanical properties compared to konjac glucomannan [9]. In this regard, a decrease in the proportion of xanthan can lead to deterioration in the mechanical properties of hydrogels.

The addition of carrot callus cells to the hydrogels based on the X:K ratio of 1:2 resulted in an increase in hardness, gumminess, elasticity, and chewiness by 2.1–2.3 times, as well as a decrease in cohesiveness and springiness by 1.1 times (Figure 5). A similar trend was observed for hydrogels with an X:K ratio of 2:1, indicating a similar effect of callus cell incorporation into the hydrogel regardless of the ratio of the two types of gums (Figure 4).

### 2.2. Rheological Properties of Gum/Callus Hydrogels

Hydrogels based on a 1.0% mixture of xanthan and konjac gums in a 2:1 (X2K1-0.1DC-J and X2K1-0.2DC-J) and 1:2 (X1K2-0.2DC-J) ratio and carrot juice with encapsulated carrot callus cells (0.1 and/or 0.2 g/mL) were chosen to study the rheological properties. Cell-free gum hydrogels (X2K1-J and X1K2-J) were used as controls. The storage modulus G′ was higher than the loss modulus G″ for all tested samples, indicating ideal elastic gel-like characteristics (Figure 6).

The storage modulus G′ and the loss modulus G″ in the entire LVE region were 1.8–2.2 and 1.4–2.4 times higher, respectively, for cell-encapsulated hydrogels (X2K1-0.1DC-J, X2K1-0.2DC-J, and X1K2-0.2DC-J) than for cell-free (X2K1-J and X1K2-J) hydrogels (Figure 6a,b). It was shown that G′ and G″ obtained as a function of frequency in the LVE region were higher for gum/callus hydrogels than for cell-free hydrogels (Figure 6c,d). This indicated that the gel network strength of cell-encapsulated hydrogels was higher than that of cell-free hydrogels. With increasing cell concentration, the G′ and G″ values increased (Figure 6a,c). The rheological analysis data are consistent with the mechanical properties analysis data, which showed an increase in the hardness of the hydrogels with the inclusion of callus cells. An increase in the values of G′, G″, and hardness with increasing cell concentration was previously shown in k-carrageenan gels enriched with thawed lupine callus tissue [20].

For all samples, low values of moduli n′ and n″ (0.012–0.149 and 0.026–0.213, respectively) and values of k″/k′ below unity (0.048–0.201) indicated the elastic nature of the obtained hydrogels (Table 1). Similar Tan[δ]_AF_ values indicated the same hydrogel spreadability. The fracture strain (γFr) values were lower for cell-encapsulated hydrogels than for cell-free hydrogels, indicating greater fragility of hydrogels with cells (Table 1). Thus, the addition of cells to the hydrogels resulted in an increase in their fragility. This is consistent with data showing a decrease in cohesiveness of hydrogels with cells compared to cell-free hydrogels, which led to the formation of more fragile gum/callus hydrogels. Previously, we showed a similar trend of increasing fragility for hydrogels based on low-methyl esterified citrus pectin with the addition of a high concentration of duckweed callus cells (0.4 g/mL) [23].

The rheological characteristics of hydrogels, such as the strength of linkage, the number of linkages, the timescale of the junction zone, and the distance of linkage, are presented in Table 1. The indices such as G′_LVE_, corresponding complex modulus (G*_FP_), limiting value of stress (τL), the frequency dependences of the elastic (k′), loss (k″), and complex (A) moduli were 1.8–2.2, 1.6, 1.6–1.7, 1,4–1.8, 2.3, and 1.5–2.0 times higher in X2K1-0.1DC-J and X2K1-0.2DC-J hydrogels with cells based on the X:K ratio of 2:1 than in X2K1-J hydrogels without cells. The slope of the loss tangent after flow point (Tan [δ]_AF_) did not change. This indicates that the inclusion of cells in the gum hydrogels increases the strength of linkage [34]. For X1K2-0.2DC-J hydrogels based on an inverted X:K ratio of 1:2 and carrot juice, the G′_LVE_ and G*_FP_ values increased by 1.9 and 2.0 times, respectively, with the addition of cells, while the Tan [δ]_AF_, τL, k′, k″, and A values did not change.

The addition of cells in hydrogels based on the X:K ratio of 2:1 led to an increase in the fracture stress (τFr) and the ratio of maximum complex modulus to linear complex modulus (G*_max_/G*_LVE_) by 1.6–1.7 and 1.1 times, respectively, indicating an increase in the number of linkages (Table 1). Parameters such as the frequency dependences of the elastic (n′), and complex (z) moduli decreased or were close. For X1K2-0.2DC-J hydrogels based on the X:K ratio of 1:2, the parameters related to the number of linkages (τFr, G*_max_/G*_LVE_, n′, and z) did not change with the inclusion of cells.

The inclusion of callus cells (0.1 and 0.2 g/mL) in hydrogels based on the X:K ratio of 2:1 caused an increase in the values of the slope of complex viscosity (η*s) by 1.8 times, indicating an increase in the timescale of junction zone (Table 1). This was probably due to an increase in the number of linkages in the hydrogels. The loss tangent (Tan[δ]_LVE_), the overall loss tangent (k″/k′), and the limiting value of strain (γL) in X2K1-0.1DC-J and X2K1-0.2DC-J hydrogels decreased or were close, respectively. The γFr values decreased with the addition of cells. The frequency dependences of the loss moduli (n″), which are characterized by the distance of linkage, tend to decrease in hydrogel samples with cells.

For X1K2-0.2DC-J hydrogels based on the X:K ratio of 1:2, the γL and γFr values decreased by 2.1 and 1.3 times, respectively, while the η*_s_, Tan[δ]_LVE_, k″/k′, and n″ values did not change with the inclusion of cells, indicating that there was no significant change in the timescale of junction zone and the distance of linkage.

A similar trend toward an increase in the strength of linkage, the number of linkages, the timescale of junction zone, and a decrease in the distance of linkage in gum/callus hydrogels based on the X:K ratio of 2:1 was previously shown for hydrogels based on low-methylesterified citrus pectin and duckweed callus cells (0.4 g/mL) [23].

In general, for X1K2-0.2DC-J and X1K2-J hydrogels based on xanthan and konjac gums with an inverted X:K ratio of 1:2 compared to hydrogels with an X:K ratio of 2:1 (X2K1-0.1DC-J, X2K1-0.2DC-J, and X2K1-J), it was observed that the strength, number, and distance of linkages did not change significantly, while the timescale of junction zone increased (Table 1).

The complex viscosity (η*_s_) of hydrogels as a function of frequency (0.03 < ω < 20.00 Hz) is presented in Figure 7. The η*_s_ values decreased with increasing frequency, which is associated with the gradual destruction of hydrogels with increasing frequency. The complex viscosity of hydrogels with an X:K ratio of 2:1 was 1.5–2.5 times higher than those with an X:K ratio of 1:2. This is probably due to the fact that by increasing the proportion of xanthan in the hydrogel, a larger number of xanthan side chains interacted with the unsubstituted regions of the glucomannan backbone of konjac [4]. This, in turn, led to a stronger synergistic interaction between xanthan and konjac macromolecules and an increase in the complex viscosity of the polysaccharide mixture. Previously, it was also shown that with an X:K ratio of 3:1, there was an increase in the viscosity of the polysaccharide mixture of xanthan and konjac compared to xanthan alone or with an X:K ratio of 1:3, which is associated with xanthan side chains responsible for special rheological behavior [9]. Gel samples containing a higher amount of xanthan gum had higher values of apparent viscosity and complex viscosity [1].

The inclusion of callus cells in hydrogels with an X:K ratio of 2:1 resulted in an increase in η*_s_ compared to the control (cell-free hydrogels) (Figure 7a). The η*_s_ values of the hydrogels increased proportionally to the increase in cell concentration. Previously, an increase in η*_s_ with increasing cell concentration was also shown in k-carrageenan gels enriched with thawed lupine callus tissue [20] and in low-methyl esterified citrus pectin hydrogels with a high concentration of duckweed callus cells (0.4 g/mL) [23]. At the same time, the addition of cells to hydrogels with an X:K ratio of 1:2 had no significant effect on the η*_s_ values (Figure 7b).

It is possible that the increase in η*_s_ values in gum/callus hydrogels with an X:K ratio of 2:1 is attributed to the emergence of a synergistic interaction of pectin macromolecules of the callus cell walls with xanthan macromolecules.

### 2.3. Sensory Properties of Gum/Callus Hydrogels

A 9-point hedonic scale was used to score the overall and consistency liking of X2K1-J and X2K1-0.2DC-J hydrogels by thirty-five volunteers. The average score of the samples was approximately 5 (“neither like nor dislike”) for both liking metrics, indicating that the inclusion of callus cells in the hydrogel did not significantly worsen its hedonic evaluation (Table 2). The aroma, taste, and perceived texture attributes, assessed using a 100 mm visual analog scale, did not differ between X2K1-J and X2K1-0.2DC-J hydrogels. Graininess was the only sensory property that differentiated between X2K1-J and X2K1-0.2DC-J hydrogels. The graininess rating of the X2K1-0.2DC-J hydrogel was 82% higher than that of the X2K1-0.2DC-J hydrogel.

The addition of cells did not affect oral processing parameters during hydrogel chewing. Electromyography (EMG) data on chewing time, number, and activity of the chewing muscles confirmed that volunteers perceived the texture of X2K1-J and X2K1-0.2DC-J hydrogels as the same. This was an unexpected result, since the EMG activity of the masseter muscle, which mediates jaw closure during mastication, is thought to correlate with the hardness of foods [35]. Perhaps the increase in hydrogel hardness resulting from cell incorporation is not large enough to alter the sensory inputs from peripheral receptors that signal the changing properties of the food during mastication. The EMG from the suprahyoid muscles that are jaw-opening muscles is hypothesized to be more sensitive to textural changes. However, the activation of the suprahyoid muscles during chewing X2K1-J and X2K1-0.2DC-J hydrogels was also identical.

The development of innovative food gel products incorporating plant tissue cultures involves the acceptability assessment, which is widely assessed using measures of liking [25,36]. First of all, overall and texture liking were measured using a nine-point hedonic scale in order to exclude the negative impact of callus cells on the overall acceptability of the gel. Adding callus cells had no effect on its hedonic ratings, suggesting that the developed gel may be a promising food gel since overall liking for a product is itself a key driver of food selection. We examined ‘hardness’, ‘springiness’, and ‘adhesiveness’ as primary, and ‘gumminess’ was scored as a secondary mechanical parameter. ‘Moisture’ and ‘easy to swallow’ subjective ratings were chosen to characterize moisture properties, which are important for characterizing the readiness of the bolus for swallowing. Graininess, as a geometric characteristic, was assumed to change when perceiving a gel containing callus cells that can form aggregates. The perception of texture properties and food acceptance also depends on the characteristics of oral processing. Several oral physiological parameters contribute directly to food oral processing, and chewing efficiency is one of the main ones. Measuring masticatory muscle activity using EMG is widely used to assess chewing efficiency and food oral processing [37].

The data obtained indicated that during sensory testing, participants did not perceive any difference in the most mechanical properties of the hydrogels. Participants did, however, notice the presence of callus cells in the hydrogel, with the X2K1-0.2DC-J hydrogel having a higher graininess rating. The fact that participants were more sensitive to graininess than other textural features underlined the importance of the food matrix’s cellular richness in addition to the mechanical properties.

## 3. Conclusions

In this study, xanthan/konjac gum hydrogels encapsulated with carrot callus cells were prepared. Hydrogels with a unique texture of artificial plant tissues were obtained. It was found that the ratio of xanthan and konjac gums and their total concentration influenced the mechanical properties of the hydrogels, such as hardness, gumminess, elasticity, and chewiness. Increasing the total gum concentration to 1.5% and the proportion of xanthan gum (X:K ratio of 3:1) in the hydrogel resulted in improved mechanical properties, which was attributed to the synergistic interactions between xanthan and konjac gum. The addition of callus cells to carrot juice-based gum hydrogels (X:K ratio of 2:1) and an increase in cell concentration from 0.1 to 0.2 g/mL led to an increase in hardness, gumminess, elasticity, and chewiness, as well as a decrease in cohesiveness. The improvement in the mechanical properties of gum hydrogels with increasing cell concentration in them was probably due to the emergence of a synergistic interaction between callus cell wall pectins and xanthan and konjac. In particular, as a result of the formation of intermolecular hydrogen bonds between the COO- groups of galacturonic acid residues of pectin and the OH groups of the main and side chains of xanthan. In addition, hydrogen bonds may be formed between the OH groups of pectin and the COO- groups of the glucuronic acid residues of the side chains and terminal side chains of xanthan. The COO- groups of pectin can form intermolecular hydrogen bonds with the OH groups on the main chain of konjac glucomannan. In addition, with increasing cell concentration in the hydrogels, more callus cell wall pectins formed intermolecular hydrogen bonds with xanthan and konjac. A decrease in the proportion of xanthan (X:K ratio of 1:2) led to a deterioration in the mechanical properties (hardness, gumminess, elasticity, and chewiness) of the hydrogels.

All hydrogel samples showed elastic gel-like behavior (G′ > G″). The mechanical properties analysis data are consistent with the rheological analysis data, which showed that the inclusion of cells in the gum hydrogels increased the strength of linkage, thereby increasing the gel network strength. In particular, the storage modulus G′_LVE_ and the loss modulus G″_LVE_ were higher for all gum/callus hydrogels than for cell-free hydrogels. The indices such as G′_LVE_, G*_FP_, τL, k′, k″, and A were higher in hydrogels (X:K ratio of 2:1) with cells than in hydrogels without cells. The addition of cells to the hydrogels based on the X:K ratio of 2:1 resulted in an increase in the number of linkages (an increase in the τFr and G*_max_/G*_LVE_ values), the timescale of junction zone, complex viscosity, and fragility (the decrease in the γFr values), and a decrease in the distance of linkage (the decrease in the n″ values). This may be due to the emergence of a synergistic interaction of pectin macromolecules of the callus cell walls with xanthan and konjac macromolecules. The rheological analysis data are consistent with the mechanical properties analysis data. In particular, the increase in rheological parameters such as G′_LVE_ and G″_LVE_, which indicate the strength of the linkage and the strength of the gel network, is consistent with the increase in the hydrogels’ hardness, which is determined by the gel network strength.

The discrepancy between measurable mechanical qualities and sensory experience of graininess can be explained by some microstructural interactions resulting from the inclusion of cells in the gel, which influence texture perception in ways that cannot be detected at perception of hardness etc. The data obtained support the widely accepted opinion that geometric features have at least as much influence as mechanical properties on how people perceive food texture.

The obtained hydrogels based on gums and immobilized carrot callus cells with unique textures may be useful for the development of diverse food textures and the production of innovative functional foods.

## 4. Materials and Methods

### 4.1. Materials

Xanthan gum (X) (viscosity 1500 mPa·s) was purchased from Xinjiang Meihua Amino Acid Co., Ltd., Wujiaqu City, China. Konjac gum (K) (viscosity 3700 mPa·s) was obtained from Foodchem International Corporation, Shanghai, China. Dried chopped carrots were taken from Stoing, St. Petersburg, Russia. Naphthylacetic acid and kinetin were purchased from Serva, Heidelberg, Germany. All other chemicals were of analytical grade.

### 4.2. Callus Culture of Daucus Carota

The callus culture of carrots *Daucus carota* subsp. sativus (Hoffm.) Arcang was obtained in the Institute of Physiology of the Federal Research Center “Komi Science Center of the Urals Branch of the Russian Academy of Sciences”. Carrot callus was cultivated at intervals of 21 days and at a temperature of 24 °C in a thermostat in the dark with a humidity of 60%. Callus was grown on modified Murashige and Skoog’s medium [38] supplemented with naphthylacetic acid (1.0 mg/L) and kinetin (0.1 mg/L). The culture conditions and hormone concentrations used were previously selected and optimized by us for the proliferation and viability of callus cells, as well as for maintaining the callus culture for a long time.

### 4.3. Preparation of Gum/Callus Hydrogels

In this work, total gum concentrations of 0.5%, 1.0%, and 1.5% were chosen because higher concentrations resulted in a significant increase in the viscosity of the polysaccharide mixture, which made the preparation of hydrogels difficult. In addition, a gum concentration of 1.0% is usually used to prepare food gels. X:K ratios of 1:1, 2:1, and 3:1 were selected based on literature data that showed the best physicochemical, rheological, textural, and sensory properties. The concentrations and X:K ratios used made it possible to obtain hydrogels with the required mechanical and rheological characteristics that were favorable for consumer evaluation.

Hydrogels were obtained based on an aqueous mixture of xanthan (X) and konjac (K) gums, at various total concentrations (0.5, 1.0, and 1.5%) and X:K ratios (1:1, 2:1, and 3:1). Xanthan gum was dissolved in distilled water overnight, and then konjac gum was added and stirred on a magnetic stirrer while heating at 75 °C for 3 h. The resulting mixture was poured into molds (size 28 × 28 mm and height 10 mm) and kept at 10 °C for 24 h. The resulting hydrogels based on an aqueous mixture of xanthan and konjac gums were studied for their mechanical properties.

To determine how the addition of callus cells at 0.1 g/mL affected the mechanical properties of the hydrogels at X:K ratios of 1:1, 2:1, and 3:1, and how the X:K ratio affected the mechanical properties of the gum/callus hydrogels, carrot callus cells (0.1 g/mL) were added to an aqueous mixture of xanthan and konjac gums at a constant total gum concentration of 1.0%.

Gum/callus hydrogels were obtained based on carrot juice. Carrot juice was prepared as follows: hot water (100 mL) was poured over dried chopped carrots (13 g) and left for 2 h, then boiled for 40 min and filtered through a mesh. Xanthan gum was dissolved in the carrot juice overnight, and then konjac gum was added and stirred on a magnetic stirrer while heating at 75 °C for 3 h. The final concentration of xanthan and konjac gums in a 2:1 or 1:2 ratio was 1.0%. Carrot callus cells were added to the resulting mixture at concentrations of 0.1 or 0.2 g/mL and mixed. The resulting mixture was poured into molds and kept as described above. As a result, hydrogel samples with carrot callus cells were obtained at a concentration of 0.1 (X2K1-0.1DC-J) and 0.2 (X2K1-0.2DC-J and X1K2-0.2DC-J) g/mL. Cell-free gum hydrogels (X2K1-J and X1K2-J) were used as controls.

### 4.4. Mechanical Properties of Gum/Callus Hydrogels

The mechanical properties of gum and gum/callus hydrogels were determined using a Texture Analyzer (TA-XT Plus, Texture Technologies Corp., Stable Micro Systems, Godalming, UK). A two-cycle compression test was performed on gel samples (10 mm high, 14 mm long, and 14 mm wide) using a P/25 cylindrical aluminum probe (25 mm diameter). The gels were compressed twice at a test speed of 1 mm/s to 100% strain at room temperature. The pre- and post-test speed was 5.0 mm/s. For different types of gels, from 8 to 26 replicates were performed. Using Texture Exponent 6.1.4.0 software (Stable Micro Systems, Godalming, UK), six parameters were extracted: hardness, cohesiveness, gumminess, elasticity, springiness, and chewiness.

### 4.5. Rheological Properties of Hydrogels

Rheological properties were investigated for hydrogels based on a 1.0% mixture of xanthan and konjac gums in a ratio of 2:1 or 1:2 and carrot juice with encapsulated carrot callus cells (0.1 and 0.2 g/mL). Cell-free gum hydrogels were used as controls. A rotational rheometer (Anton Paar, Physica MCR 302, Graz, Austria) with parallel plate geometry (diameter 25 mm; gap 4.0 mm) was used for the rheological measurements. A controlled shear rate mode at 20 °C with a constant frequency (1 Hz) and stress (9.0 Pa) was used. Strain sweep evaluation was performed for strain amplitudes ranging from 0.01 to 100%. The G′_LVE_, G″_LVE_, G*_FP_, Tan [δ]_AF_, τL, G*_max_/G*_LVE_, τFr, Tan [δ]_LVE_, γL, and γFr were evaluated according to [22,34,39,40,41,42]. The frequency dependences of k′, k″, k″/k′, n′, n″, A, z, and η*_s_ were determined as described in [43].

### 4.6. Sensory Properties of Gum/Callus Hydrogels

Thirty-five volunteers of both sexes without masticatory or swallowing dysfunctions were involved in the research. The invitation to participate was distributed to scientific staff and university students by announcement and e-mail. Inclusion criteria were aged from 18 to 45 years, male and female, and the desire to participate in the study was confirmed by informed consent. Exclusion criteria were acute inflammatory diseases and dental treatments. All participants were given detailed instructions on how to conduct the survey. They then signed informed consent and completed a participant questionnaire in which they indicated their gender, age, height, weight, and whether they visited a dentist in the last week. The study protocol was approved by the Bioethics Committee of the Institute Physiology of the Federal Research Centre “Komi Science Center of the Urals Branch of the Russian Academy of Sciences” (approval no. 10/10 March 2022). The participants underwent a single test lasting 30 min (from 11:00 to 13:00) in individual booths of a sensory room under standard light exposure and temperature (20 °C). Three test sessions were performed by each subject: (1) hedonic evaluation; (2) sensory score assessment; and (3) EMG recording under unilateral chewing as described earlier [22]. First, the participants were asked to score samples from extremely disliked (1 point) to extremely liked (9 points). The three samples were served in white plastic containers: 2% agar hydrogel (a training sample; it was not taken into account), X2K1-J, and X2K1-0.2DC-J hydrogels. Then, participants evaluated nine sensory and texture attributes using a 100 mm visual analog scale.

In the third session, EMG activity from the masseter (Masseter muscle), temporal muscles (Temporalis muscle), and suprahyoid muscles (Suprahyoid muscles) was determined by surface EMG recording using a Neuro-MEP system with 4-channel amplifiers (Neurosoft, Ivanovo, Russia), with subsequent visualization and data analysis in the Neuro-MEP.NET program (ver. 4.2.6.5). For this purpose, three pairs of disposable gel electrodes (F9040, Fiab, Vicchio, Italy) were attached to the left or right muscles depending on the preferred chewing side as described earlier [35,44]. Electrodes were attached to the anterior temporal and masseter muscles, which close the jaw, and to the suprahyoid muscles, which open the jaw. The masseter and temporal muscles were located by palpation while the participants clenched their teeth. Two pairs of electrodes were then attached as close as possible to the center of the muscle belly, parallel to the muscle. The first pair of electrodes was attached to a line connecting the lower part of the ear with the corner of the mouth. The second pair of electrodes was attached to a line connecting the corner of the eye and the upper part of the ear. The third pair of electrodes was positioned perpendicular to a line connecting the middle of the chin and the neck. The electrodes (11 × 34 mm) were placed on the skin strictly symmetrically with an inter-electrode distance of 20 mm and fixed with a fixation patch. The skin was degreased at the attachment site to reduce resistance. The ground band electrode was applied to the wrist of the left hand. Before EMG recording, participants were instructed to chew freely until it was easy to swallow.

### 4.7. Statistical Analysis

For the study of mechanical characteristics, the significance of differences between the mean values was estimated using Student’s *t*-test. A one-way ANOVA was used to determine differences in the rheological characteristics of hydrogels. Statistical differences were considered significant when *p*-values were lower than 0.05. The data obtained are presented as mean ± standard deviation (S.D.). Normality of data distribution was determined by the Shapiro–Wilk test. Since the data had a non-normal distribution, the Friedman’s test and the Durbin post hoc test were used to compare the mean values of sensory and EMG data obtained during the chewing of different food gels.

## Figures and Tables

**Figure 1 gels-10-00771-f001:**
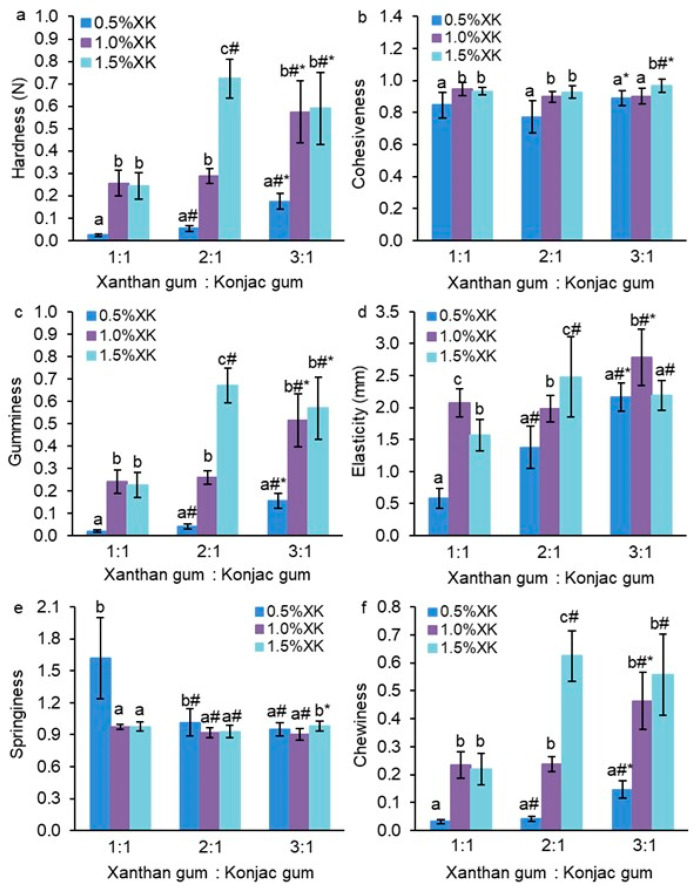
Нardness (**a**), cohesiveness (**b**), gumminess (**c**), elasticity (**d**), springiness (**e**), and chewiness (**f**) of hydrogels based on an aqueous mixture of xanthan (X) and konjac (K) gums. The data are presented as the mean ± S.D., *n* = 12. Different lowercase letters (a, b, and c) indicate significant differences (*p* < 0.05) between the means for different gum concentrations; # *p* < 0.05 vs. X:K ratio of 1:1, * *p* < 0.05 vs. X:K ratio of 2:1.

**Figure 2 gels-10-00771-f002:**
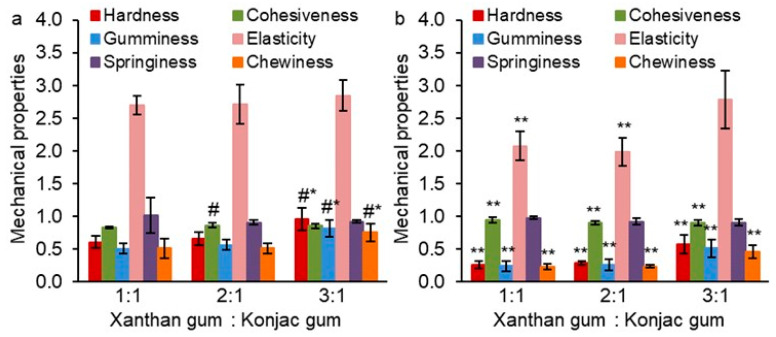
Mechanical properties of gum/callus hydrogels based on a 1.0% aqueous mixture of xanthan and konjac gums in different ratios (1:1, 2:1, and 3:1) and 0.1 g/mL carrot callus cells (**a**). Cell-free gum hydrogels were used as controls (**b**). Hardness and elasticity are expressed in units of H and mm, respectively. The data are presented as the mean ± S.D., *n* = 8. # *p* < 0.05 vs. X:K ratio of 1:1, * *p* < 0.05 vs. X:K ratio of 2:1, ** *p* < 0.05 vs. corresponding experimental characteristics in (**a**).

**Figure 3 gels-10-00771-f003:**
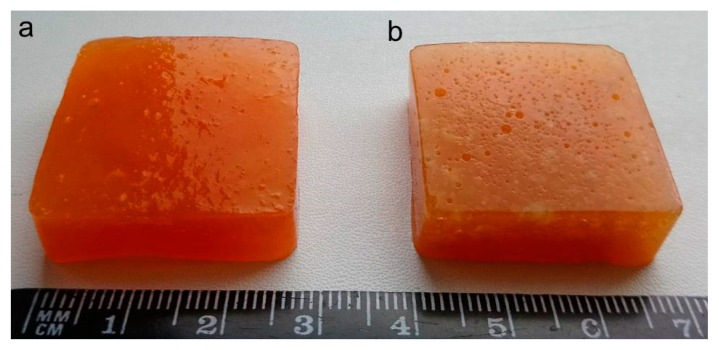
Digital images of cell-free gum hydrogels (X2K1-J) (**a**) and gum/callus hydrogels based on carrot juice, a 1.0% mixture of xanthan and konjac gums in a 2:1 ratio, and carrot callus cells (0.2 g/mL) (X2K1-0.2DC-J) (**b**).

**Figure 4 gels-10-00771-f004:**
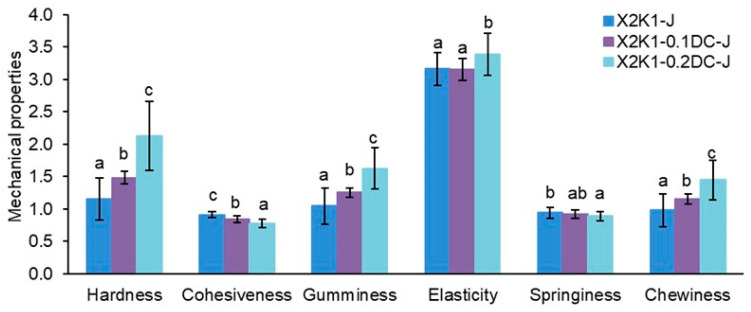
The effect of carrot callus cell concentration (0.1 and 0.2 g/mL) on the mechanical properties of hydrogels based on carrot juice and a 1.0% mixture of xanthan and konjac gums in a 2:1 ratio. Hardness and elasticity are expressed in units of N and mm, respectively. The data are presented as the mean ± S.D., *n* = 40. Different lowercase letters (a, b, and c) indicate significant differences (*p* < 0.05) between the means for different callus cell concentrations. Cell-free gum hydrogels (X2K1-J) were used as controls.

**Figure 5 gels-10-00771-f005:**
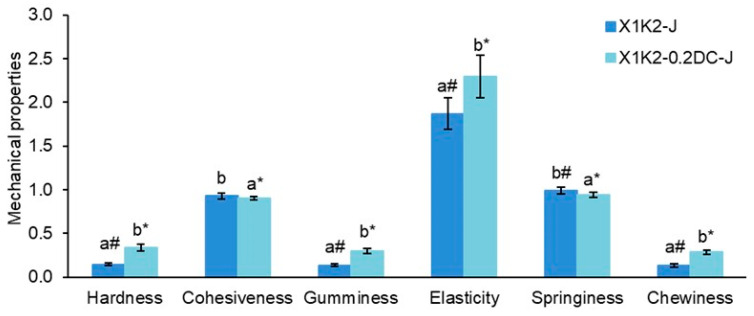
Mechanical properties of hydrogels based on a 1.0% mixture of xanthan and konjac gums in a 1:2 ratio, 0.2 g/mL carrot callus cells, and carrot juice. Hardness and elasticity are expressed in units of H and mm, respectively. The data are presented as the mean ± S.D., *n* = 12. Different lowercase letters (a and b) indicate significant differences (*p* < 0.05) between the means. # *p* < 0.05 vs. X2K1-J (X:K ratio of 2:1) in Figure 4, * *p* < 0.05 vs. X2K1-0.2DC-J (X:K ratio of 2:1) in Figure 4. Cell-free gum hydrogels (X1K2-J) were used as controls.

**Figure 6 gels-10-00771-f006:**
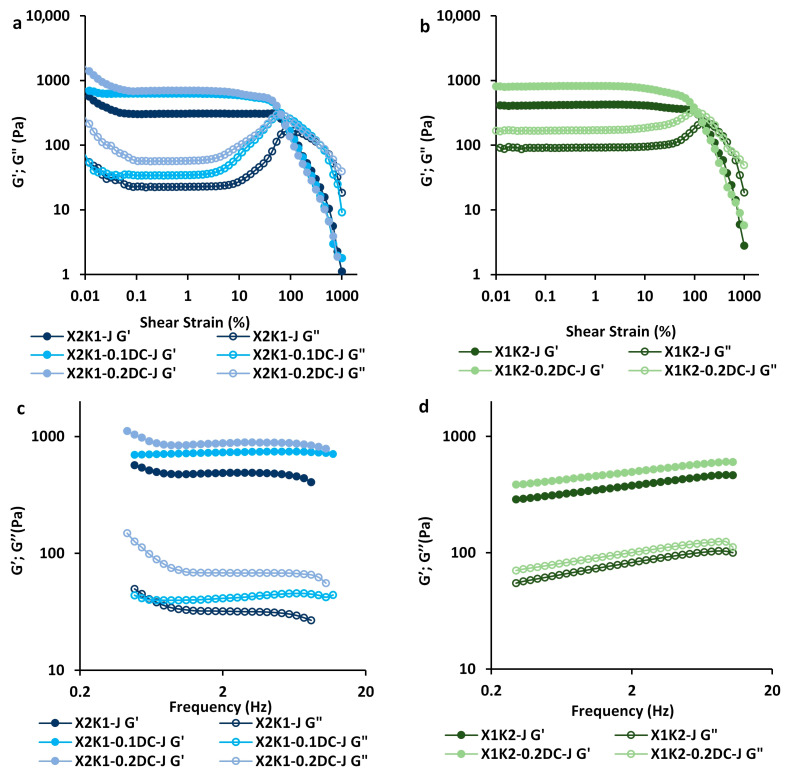
Rheological properties of hydrogels based on 0.1 and 0.2 g/mL carrot callus cells, carrot juice, and a 1.0% mixture of xanthan and konjac gums in a 2:1 (**a**,**c**) and 1:2 (**b**,**d**) ratio. Storage modulus (G′, filled symbols) and loss modulus (G″, empty symbols) are represented as a function of shear strain (**a**,**b**) or frequency (**c**,**d**). Cell-free gum hydrogels (X2K1-J, X1K2-J) were used as controls.

**Figure 7 gels-10-00771-f007:**
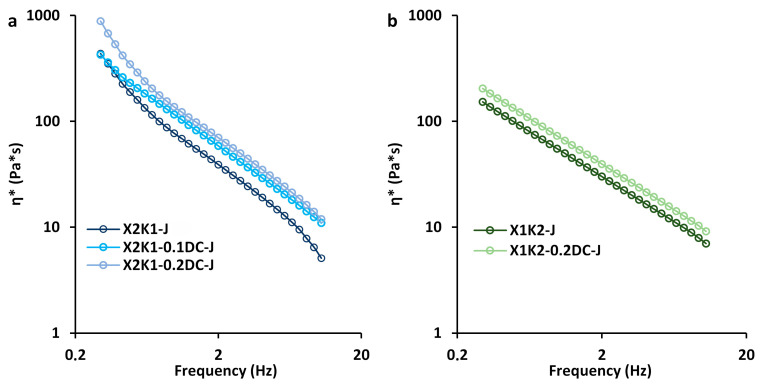
Complex viscosity as a function of frequency of hydrogels based on 0.1 and 0.2 g/mL carrot callus cells, carrot juice, and a 1.0% mixture of xanthan and konjac gums in a 2:1 (**a**) and 1:2 (**b**) ratio. Cell-free gum hydrogels (X2K1-J, X1K2-J) were used as controls.

**Table 1 gels-10-00771-t001:** The rheological properties of gum/callus hydrogels (X2K1-0.1DC-J and X2K1-0.2DC-J) and cell-free gum hydrogels (X2K1-J).

Parameters	X2K1-J (Control) (*n* = 8)	X2K1-0.1DC-J (*n* = 8)	X2K1-0.2DC-J (*n* = 8)	X1K2-J (Control) (*n* = 8)	X1K2-0.2DC-J (*n* = 8)
	Strength of linkage				
G′_LVE_ (Pa)	344.5 ± 1.8 ^a^*	613.7 ± 28.5 ^b^	773.4 ± 17.4 ^cα^	417.1 ± 7.0 ^1#^	806.6 ± 27.0 ^2β^
G*_FP_ (Pa)	353.9 ± 24.5 ^a^*	550.9 ± 44.4 ^b^	548.2 ± 75.3 ^bα^	405.3 ± 11.1 ^1^*	810.6 ± 473.1 ^2α^
Tan [δ]_AF_	0.015 ± 0.005 ^a^*	0.018 ± 0.004 ^a^	0.018 ± 0.009 ^aα^	0.006 ± 0.001 ^1^*	0.007 ± 0.001^1α^
τL (Pa)	347.6 ± 20.4 ^a^*	599.1 ± 53.4 ^c^	563.3 ± 37.1 ^bα^	368.6 ± 104.0 ^1^*	629.3 ± 299.8 ^1α^
k′ (Pa·s)	512.2 ± 94.4 ^a^*	725.1 ± 85.6 ^b^	908.7 ± 157.2 ^cα^	366.1 ± 18.4 ^1#^	400.8 ± 25.4 ^1β^
k″(Pa·s)	39.9 ± 14.8 ^a^*	41.1 ± 14.6 ^a^	92.4 ± 14.9 ^bα^	73.7 ± 2.9 ^1#^	73.0 ± 3.9 ^1α^
A (Pa·s)	489.3 ± 100.3 ^a^*	716.3 ± 90.0 ^b^	960.5 ± 155.0 ^cα^	366.0 ± 18.2 ^1^*	400.7 ± 25.6 ^1β^
	Number of linkages				
G*_max_/G*_LVE_	1.002 ± 0.008 ^a^*	1.027 ± 0.008 ^b^	1.063 ± 0.012 ^cα^	1.028 ± 0.014 ^1^*	1.013 ± 0.004 ^1β^
τFr (Pa)	177.7 ± 27.4 ^a^*	283.7 ± 31.8 ^b^	302.6 ± 52.3 ^bα^	215.6 ± 48.5 ^1^*	311.7 ± 130.2 ^1α^
n′	0.096 ± 0.058 ^b^*	0.012 ± 0.008 ^a^	0.037 ± 0.024 ^aα^	0.149 ± 0.003 ^1^*	0.133 ± 0.008 ^1β^
z	41.5 ± 4.2 ^a^*	52.1 ± 25.6 ^a^	44.0 ± 34.0 ^aα^	7.2 ± 0.9 ^1#^	7.4 ± 0.4 ^1α^
	Timescale of junction zone				
Tan [δ]_LVE_	0.077 ± 0.028 ^b^*	0.057 ± 0.006 ^a^	0.081 ± 0.015 ^bα^	0.220 ± 0.007 ^1#^	0.194 ± 0.020 ^1β^
k″/k′	0.085 ± 0.011 ^a^*	0.048 ± 0.021 ^a^	0.094 ± 0.001 ^bα^	0.201 ± 0.011^1#^	0.182 ± 0.003 ^1β^
η*_s_ (Pa·s)	85.2 ± 15.7 ^a^*	115.9 ± 13.8 ^b^	156.6 ± 26.6 ^cα^	56.8 ± 3.5 ^1#^	64.3 ± 4.5 ^1β^
γL (%)	49.9 ± 7.6 ^b^*	11.6 ± 2.9 ^a^	43.1 ± 5.4 ^bα^	75.1 ± 6.9 ^1#^	36.1 ± 8.9 ^2α^
γFr (%)	100.1 ± 22.8 ^b^*	68.8 ± 9.3 ^a^	68.7 ± 9.3 ^aα^	149.9 ± 26.8 ^1#^	112.8 ± 8.2 ^2β^
	Distance of linkage				
n″	0.213 ± 0.103 ^b^*	0.026 ± 0.021 ^a^	0.154 ± 0.068 ^bα^	0.196 ± 0.005 ^1^*	0.167 ± 0.026 ^1α^

The data are presented as the mean ± S.D., *n* = 4 and 8. Differences are significant (*p* < 0.05) between means labeled with different lowercase letters (a, b, and c). ^1,2^ *p* < 0.05 between X1K2-J and X1K2-0.2DC-J, *^,#^ *p* < 0.05 between X2K1-J and X1K2-J, ^α,β^ *p* < 0.05 between X2K1-0.2DC-J and X1K2-0.2DC-J. The storage modulus (G′_LVE_), corresponding complex modulus (G*_FP_) with the stress at flow point, the slope of the loss tangent after flow point (Tan [δ]_AF_), limiting value of stress (τL), G*_max_/G*_LVE_ is the ratio of maximum complex modulus to linear complex modulus, the fracture stress (τFr), the loss tangent (Tan [δ]_LVE_), limiting value of strain (γL), and the fracture strain (γFr). The frequency dependences of the elastic (k′ and n′), loss (k″ and n″, and complex (A and z) moduli; overall loss tangent (k″/k′); and the slope of complex viscosity (η*_s_). Cell-free gum hydrogels (X2K1-J) were used as controls.

**Table 2 gels-10-00771-t002:** Liking and sensory evaluations, and oral processing parameters for X2K1-J and X2K1-0.2DC-J hydrogels.

Parameter	Hydrogel	
	X2K1-J	X2K1-0.2DC-J
Nine-point hedonic scale		
Overall liking	5.4 ± 1.7 ^a^	4.9 ± 1.6 ^a^
Consistency liking	5.1 ± 1.7 ^a^	4.9 ± 1.6 ^a^
100 mm visual analog scale		
Aroma	42 ± 25 ^a^	41 ± 26 ^a^
Taste	56 ± 19 ^a^	51 ± 17 ^a^
Hardness	26 ± 21 ^a^	24 ± 18 ^a^
Springiness	58 ± 27 ^a^	50 ± 25 ^a^
Adhesiveness	20 ± 20 ^a^	23 ± 22 ^a^
Gumminess	28 ± 20 ^a^	26 ± 19 ^a^
Moisture	52 ± 23 ^a^	53 ± 28 ^a^
Easy to swallow	82 ± 17 ^a^	80 ± 19 ^a^
Graininess	31 ± 29 ^a^	57 ± 33 ^b^
EMG parameters		
Chewing time, s	19 ± 9 ^a^	19 ± 8 ^a^
Number of chews	25 ± 11 ^a^	25 ± 11 ^a^
Activity of Masseter muscle, mV × s	0.19 ± 0.14 ^a^	0.19 ± 0.10 ^a^
Activity of Temporalis muscle, mV × s	0.35 ± 0.20 ^a^	0.19 ± 0.12 ^a^
Activity of Suprahyoid muscles, mV × s	0.59 ± 0.29 ^a^	0.40 ± 0.21 ^a^

The data are presented as the mean ± S.D., *n* = 35. Differences are significant (*p* < 0.05) between means labeled with different lowercase letters (a and b).

## Data Availability

The data that support the findings of this manuscript are available from the corresponding author upon reasonable request.

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
