# Peer review of "Effect of Carrot Callus Cells on the Mechanical, Rheological, and Sensory Properties of Hydrogels Based on Xanthan and Konjac Gums"

_gels, 2024, doi:10.3390/gels10120771_

Round 1

Reviewer 1 Report

Comments and Suggestions for Authors

Hi dear Editorial board and the respected authors

This article "Effect of carrot callus cells on the mechanical, rheological, and sensory properties of hydrogels based on xanthan and konjac gums” was revised and has a novelty and I recommend it for publication after consideration of the following comments.

Abstract:

·         Abstract: The aim of the present work was to identify the effect of the inclusion of carrot callus cells in hydrogels based on xanthan and konjac gums on the mechanical, rheological, and sensory properties of the final product. Please rewrite because it is the same of title. Please introduce for carrot callus cells.

·         The type of statistical design used in this research including the name of treatments should be mentioned.

·         Inclusion and future study include in the end of the abstract.

·         What particular biochemical interactions occur between carrot callus cell wall pectin macromolecules and xanthan and konjac gum macromolecules that lead to the observed synergistic effects on hydrogel mechanical properties?

·         How can different concentrations of carrot callus cells (0.1 g/mL vs. 0.2 g/mL) impact mechanical qualities including hardness, gumminess, and elasticity? Are there specific thresholds beyond which these properties plateau or decline?

·         Can you elaborate on the methodology used for the sensory evaluation? How were the sensory attributes (e.g., graininess, hedonic ratings) quantified, and what statistical analyses were performed to determine the significance of the effects observed with the inclusion of callus cells?

Introduction:

·         What chemical interactions between xanthan gum and konjac glucomannan help to produce the observed synergistic gelling effect? Could you explain on the many forms of bonding (e.g., hydrogen bonds, van der Waals forces) that help the hydrogel establish its three-dimensional grid structure?

·         How does changing the concentration of carrot callus cells in the hydrogel impact its mechanical and rheological properties? Are there precise thresholds or ideal doses at which these qualities markedly improve or degrade?

·         Can you elaborate on the sensory evaluation methods used to determine the textural qualities of the hydrogels? What criteria were utilized to evaluate sensory qualities such as graininess and general acceptability, and how were they statistically assessed?

Materials and methods:

·         Materials and methods section ought to insert after introduction section.

·                       Please write materials as Company Name (City, Country), especially for chemical analysis assessment which used in the study.

·         Could you please offer additional information on the particular environmental conditions (e.g., light, humidity) used to cultivate carrot callus cells on the modified Murashige and Skoog medium? Furthermore, how may these variables impact the proliferation and vitality of callus cells?

·         What particular factors influenced the choice of X:K ratios (1:1, 2:1, and 3:1) and total concentrations (0.5%, 1.0%, and 1.5%) for hydrogel formulations? How do these parameters relate to the required mechanical and rheological characteristics of the resulting hydrogels?

·         Could you please explain the reasoning behind the selection of sensory evaluation methods, namely the usage of hedonic scoring and EMG recording? How do these strategies aid to knowing the consumers?

Results:

·         All Tables and Figures: The alphabetical statistical letters for the means should all be modified such that the greatest number has the letter a and as the numbers go lower, letters b, c etc.

·          

Discussion:

·            Discussion text must grammar improve and in some cases it is very weak and maybe there is no discussion at all.

Conclusions:

Conclusion is very general, try to make it more scientific, comprehensive and concise in detail, especially.

·      According to the sensory evaluation, participants did not notice significant changes in the majority of mechanical qualities of the hydrogels, however there was a difference in graininess. How do you explain the apparent discrepancy between measurable mechanical qualities (such as hardness and gumminess) and sensory experience of graininess? Could there be microstructural interactions that impact texture perception in ways that cannot be detected mechanically?

·      The study found that increasing the concentration of carrot callus cells improved the mechanical characteristics of the hydrogels. What precise processes or structural changes occur within the hydrogels to achieve these improvements? How do these changes differ between the two examined callus cell concentrations (0.1 g/mL vs. vs. 0.2 g/mL)?

·      According to the rheological study, the addition of callus cells boosted gel network strength and impacted the hydrogels' complicated viscosity. Can you describe the exact rheological parameters assessed and how they relate to the reported mechanical properties? Furthermore, how can these findings influence the development of future hydrogels for various uses in the food industry?

Reviewer 2 Report

Comments and Suggestions for Authors

Interesting paper. Legends of all figures are not clear. For example what is the difference between fig. 4 and 5? Both figures discuss about mechanical properties.

Figure 4. The effect of carrot callus cells (0.1 and 0.2 g/mL) on the mechanical properties of hydro- gels based on carrot juice and a 1.0% mixture of xanthan and konjac gums in a 2:1 ratio. Hardness and elasticity are expressed in units of H and mm, respectively. I know that hardness is a strength in units of N.

I believe it would be more clear to differentiate the samples in the materials and methods. The authors use 2 different concentrations right? 0.1 and 0.2 g/ml.

Please explain EMG data and describe the superficial masseter, anterior temporalis, and su-prahyoid muscles was monitored using bipolar electrodes (11 x 25 mm) that were sepa-rated from each other by approximately 20 mm.

Who did this? Was this carried out in a dentistry lab or what?

Regarding this set of experiments and sensory evaluation an ethical approval is required.

Comments on the Quality of English Language

English needs to be checked again

Round 2

Reviewer 2 Report

Comments and Suggestions for Authors

Authors have replied sufficiently and paper can be accepted as it is.